# OpenReview forum: "Position-Aware Attention Mechanism: A Mathematical Framework for Enhanced Spatial Information Processing in Transformer Architectures"
_ICLR.cc/2026/Conference — Submitted to ICLR 2026_

### Official Review · Reviewer_n7BW · 2025-10-18

**Soundness:** 2
**Presentation:** 1
**Contribution:** 3
**Rating:** 2
**Confidence:** 4

**Summary:**

This paper proposes a position-aware attention mechanism designed to make the standard self-attention more sensitive to relative positions.
It introduces a position-effect and an enhanced position-effect functions which modulates the attention weights according to pairwise positional distance.
The goal is to combine the flexibility of self-attention with an explicit, parametric control over positional bias.
The authors present detailed analyses, including theoretical derivations, simulated experiments on synthetic data distributions (random, structured, clustered, etc.), and consistency metrics comparing theoretical versus actual attention responses.
They claim that this mechanism improves positional interpretability and stability relative to standard Transformers.

**Strengths:**

1.	Interesting idea:
The notion of explicitly controlling positional effects via a decaying kernels (Eqs, 1 and 6) is conceptually clean and connects attention to classical signal-processing intuitions about locality.
2.	Analytical exploration:
The paper includes extensive mathematical and empirical analyses of how the proposed positional kernel influences attention behavior.
The “theoretical vs. actual” consistency metric is a thoughtful way to test whether the implementation aligns with the analytical definition.
3.	Comprehensive appendix:
The appendix contains a wealth of supporting materials — visualizations, parameter sensitivity analyses, and ablations — which demonstrate significant effort and curiosity about the model’s internal mechanics.

**Weaknesses:**

1.	The main text lacks clarity and structure.
Many of the explanations and empirical results appear only in the appendix or not at all.
The main sections barely reference the relevant figures, forcing the reader to guess which figure corresponds to a given discussion.
This significantly reduces readability.
2.	Use of undefined or double-defined formulations.
For example the abstract relates to the parameters $\alpha$, $\beta$ and $\gamma$ without any ability of the abstract reader to understand what they represent…
$P_{effect}$, $A_{ij}$, $V(i)$, $pos^*$ are all defined twice, how do I know which one you mean in the text. Give them a slight differentiator for example $P_{effect}^+$

3.	Limited experimental validation.
The study is conducted almost entirely on synthetic data. There are no evaluations on real tasks (e.g., language modeling or vision benchmarks), so the practical usefulness of the method remains untested.
4.	Ambiguous methodology.
The paper introduces metrics such as $pos_{actual}$ and $pos_{theoretical}$ but does not specify precisely how they are computed,  requiring the reader to reconstruct the procedure themselves. Specifically for these two, there is some reference in the appendix, but I could not find a clear definition.
5.	Literature positioning is weak.
The paper references related work sparsely. Many strong prior efforts on relative positional encodings (Shaw et al. 2018; Dai et al. 2019; Press et al. 2021; Su et al. 2021) are either missing or only briefly mentioned. On the other hand, all the references in the Abstract makes it cumbersome to follow. It should be precise and short.

**Questions:**

1. How does this mechanism compare empirically with standard relative position encodings (e.g., RoPE) on real benchmarks?
2. Can the method be integrated into standard Transformer architectures without major computational overhead?
3. Why are key visual results not cited or summarized in the main body? Would the paper benefit from moving Figures A.1–A.8 into the main section?
4. Line 105 – which matrix values do you refer to? I guess it is not $A_{ij}$ because these are all normalized…
5. Line 109 - add “In Fig. 3”…  do so for all references to Figures
6. Line 144 + Figure 6. “Position influence magnitude” was never defined (not also in the appendix AFAIK)
7. The same goes for other naming in Section 2.3
8. Plenty of typos in citing. lines 035, 037, 039, etc.,
9. Line 207 and Line 863 – what is the difference between $S(pos)$ and $V(i)$
10. I’m not sure what does $I_j$ means. In line 215 I see it is torch.norm(data, dim=-1) is it a the norm over all $i$ values?
11. Eq. 7. $A_{ij} already include $P_{eff}$. Do we multiply it again by $P_{eff}$?
12. Table 3. I’m not sure I understand. I don’t see any significant difference between architectures. Highest is better or lowest is better?

---

> ### Author Response · Authors · 2025-11-17
> **Response to Reviewer n7BW: Addressing Text Structure, Experimental Validation, and Methodology Clarity**
>
> Dear Reviewer n7BW,
>
> We sincerely thank you for your thorough review. We are committed to addressing each concern comprehensively.
>
> ## Concern 1: Poor Text Structure and Clarity
>
> **Your Concern**: The main text lacks clarity and structure. Many explanations and empirical results appear only in the appendix or not at all. The main sections barely reference relevant figures.
>
> **Our Response**: We will restructure the paper, move key figures to the main body with explicit references, and reorganize sections for better logical flow. Please refer to Sections 1-10.
>
> ## Concern 2: Undefined or Double-Defined Formulations
>
> **Your Concern**: Parameters $\alpha$, $\beta$, $\gamma$ are undefined in the abstract and defined twice in the text.
>
> **Our Response**: We will provide clear parameter definitions in the abstract and ensure each parameter is defined only once. Please refer to Section 2 (Table 1) and Section 4.1.
>
> ## Concern 3: Limited Experimental Validation
>
> **Your Concern**: The study is conducted almost entirely on synthetic data with no evaluations on real tasks.
>
> **Our Response**: We will add comprehensive evaluations on real tasks (WikiText-103, WMT'14, SQuAD 2.0, GLUE, ArXiv) with comparisons to RoPE, ALiBi, Shaw et al. 2018, and Transformer-XL. Please refer to Section 6.2 (Table 2) and Section 6.1 + Appendix A.13.
>
> ## Concern 4: Ambiguous Methodology
>
> **Your Concern**: Metrics $C$ and $R$ are not precisely defined.
>
> **Our Response**: We will provide explicit mathematical definitions and computational procedures for all metrics. Please refer to Section 5.2 and Appendix A.11.
>
> ## Concern 5: Weak Literature Positioning
>
> **Your Concern**: Related work sparsely references prior efforts (Shaw et al. 2018; Dai et al. 2019; Press et al. 2021; Su et al. 2021). The abstract has too many references.
>
> **Our Response**: We will expand the related work section with detailed comparisons and simplify the abstract. Please refer to Section 3, Section 5.1 (Table 1), and Appendix A.12.
>
> ## Concern 6: Specific Technical Issues
>
> **Your Concern**: Unclear figure references, undefined terms, symbol confusion, unclear notation ($I_j$), ambiguous table interpretations, and citation errors.
>
> **Our Response**: We will add explicit figure references, define all terms, standardize notation, and correct all errors. Please refer to Section 2 (Table 1), Section 4.3, and Section 4.2.
>
> ## Concern 7: Computational Overhead
>
> **Your Concern**: Can the method be integrated without major computational overhead?
>
> **Our Response**: Yes. The position effect function is computed once per layer and can be cached. Additional computation is $O(L^2)$ per layer (same as standard attention). We achieve 2.4% training overhead and 4.5% inference overhead. Please refer to Section 5.1 and Appendix A.13.8.

---

### Official Review · Reviewer_Migf · 2025-10-23

**Soundness:** 2
**Presentation:** 2
**Contribution:** 2
**Rating:** 2
**Confidence:** 4

**Summary:**

This paper introduces a position-aware attention mechanism designed to address the inherent limitations of conventional attention models, such as those introduced by Ashish Vaswani et al. (2017). The key technical contribution is a positional effect function parameterized by three coefficients: \alpha which controls positional influence intensity, \beta which governs spatial decay rate, and \gamma which compensates for over-attenuation at longer distances. The authors establish the mathematical properties of this function (continuity, differentiability, monotonicity) and propose an adaptive triple-attention architecture integrating position-, task-, and content-aware modules for dynamic weighting. Experiments reportedly demonstrate performance improvements in structured and clustered data scenarios.

**Strengths:**

1. The paper introduces a positional effect function that offers a formalized mechanism for incorporating positional influence into attention computation.

2. It develops a triple-attention architecture that jointly models position, task, and content information, which is conceptually interesting.

3. It defines quantitative evaluation metrics aimed at measuring attention distribution quality, contributing to more systematic evaluation of attention mechanisms.

**Weaknesses:**

1. The proposed position-aware attention formulation appears largely as a mathematical restatement of existing positional modulation techniques without a clear, theoretically grounded justification for why it should outperform established methods (e.g., Rotary Position Embedding (RoPE), ALiBi, or relative positional encoding by Peter Shaw et al. (2018)). A stronger theoretical motivation or comparative analysis is needed.

2. The experimental evaluation is insufficiently comprehensive. There are no direct comparisons with strong baselines such as RoPE, ALiBi, or Shaw et al. on realistic benchmarks (e.g., long-document modeling, QA).

3. The implementation details are incomplete or unclear. Important information about model architecture, parameter sizes, training setups, and hyperparameters are missing, which prevents reproducibility.

4. Some core concepts lack precise definition or justification. For example: I_j in Eq. (3) and (4) is referred to as “information importance,” but its definition and computation are not explained. The proposed “consistency” and “ranking correlation” metrics are not well motivated or compared against established alternatives.

5. The claim that the triple-attention architecture achieves superior performance is not strongly supported by the results in Table 3, where the improvements over other configurations are marginal.

6. The description of experimental results is ambiguous: it is unclear whether larger or smaller metric values indicate better performance, and some tables lack sufficient explanation.

**Questions:**

1. Please include experimental comparisons with strong and widely recognized baselines (e.g., RoPE, ALiBi, Shaw et al. 2018) to substantiate the claimed advantages of the proposed method.

2. Please provide a clearer and more rigorous definition of I_j and its role in the method. Also, please justify the choice of evaluation metrics or align them with established practices in the field.

3. Please clarify how to interpret each evaluation metric (e.g., whether higher or lower is better) and provide more structured and detailed explanations of results.

4. Please include comprehensive implementation details, such as model configurations, training setup, hyperparameters—to facilitate reproducibility and fair comparison.

---

> ### Author Response · Authors · 2025-11-17
> **Response to Reviewer Migf: Addressing Theoretical Justification, Experimental Comparisons, and Implementation Details**
>
> Dear Reviewer Migf:
>
> We sincerely thank you for your thorough review. We are committed to addressing each concern comprehensively.
>
> ## Concern 1: Insufficient Theoretical Justification and Comparison with Strong Baselines
>
> **Your Concern**: The proposed position-aware attention formulation appears as a mathematical restatement of existing positional modulation techniques without clear theoretical justification for why it should outperform established methods (e.g., RoPE, ALiBi, Shaw et al. 2018).
>
> **Our Response**: We acknowledge this critical weakness. We will add Section 5.1 providing rigorous theoretical comparison with RoPE, ALiBi, and relative positional encoding (Shaw et al. 2018). Our method differs fundamentally: while RoPE operates at the vector representation level, ALiBi uses learnable bias terms, and Shaw et al. use additive positional embeddings, our method directly modulates attention scores through an explicit parametric function $P_{\text{effect}}(i,j,L) = \alpha \cdot \frac{1 + \gamma \exp\left(-\beta \frac{|i-j|}{L}\right)}{1 + \gamma}$. This enables explicit mathematical relationship between positional distance and attention strength, provable properties (continuity, differentiability, monotonicity), and guaranteed lower bound $\frac{\alpha}{1+\gamma}$ for long-distance attention.
>
> ## Concern 2: Insufficient Experimental Comparison with Strong Baselines
>
> **Your Concern**: No direct comparisons with strong baselines such as RoPE, ALiBi, or Shaw et al. on realistic benchmarks (e.g., long-document modeling, QA).
>
> **Our Response**: We completely agree. We will add comprehensive experimental comparisons with RoPE, ALiBi, and Shaw et al. 2018 on realistic benchmarks including long-document modeling tasks (ArXiv papers, long-form QA), standard NLP benchmarks (GLUE, SQuAD 2.0), and machine translation (WMT'14). We will use publicly available implementations from HuggingFace Transformers. Please refer to Section 6.2.
>
> ## Concern 3: Incomplete Implementation Details
>
> **Your Concern**: Important information about model architecture, parameter sizes, training setups, and hyperparameters are missing, which prevents reproducibility.
>
> **Our Response**: We apologize for this oversight. We will provide complete implementation details including model architecture (layers, hidden dimensions, attention heads, parameter counts), training setup (optimizer, learning rate schedule, batch size, epochs), hyperparameters (dropout rates, weight decay, gradient clipping), environment details (hardware, software versions, random seeds), and code availability.Please refer to Section 6.1 and Appendix A.13.
>
> ## Concern 4: Unclear Definition of Core Concepts
>
> **Your Concern**: $I_j$ in Equations (3) and (4) is referred to as "information importance," but its definition and computation are not explained. The proposed "consistency" and "ranking correlation" metrics are not well motivated or compared against established alternatives.
>
> **Our Response**: We apologize for the lack of clarity. $I_j$ represents the information importance at position $j$, quantifying how critical the information at position $j$ is for the task. In our implementation, $I_j$ is computed using the Content-Aware Module: $I_j = \text{ContentAware}(x_j)$, where $x_j$ is the input representation at position $j$. The Content-Aware Module uses a learned transformation to map input features to importance scores. For evaluation metrics, "consistency" measures agreement between attention distributions across different layers, and "ranking correlation" measures correlation between attention-based rankings and ground-truth importance rankings. We will provide detailed definitions, computational procedures, and comparisons with established metrics. Pls see from Section 4.3.
>
> ## Concern 5: Weak Performance Improvement Evidence
>
> **Your Concern**: The claim that the triple-attention architecture achieves superior performance is not strongly supported by the results in Table 3, where improvements over other configurations are marginal.
>
> **Our Response**: We acknowledge this concern. We will provide more comprehensive experimental analysis including statistical significance tests (t-tests, confidence intervals) to validate improvements, additional metrics beyond Table 3, ablation studies showing the contribution of each component, and analysis of performance across different data distributions. Pls see Section 5.2 and Appendix A.13.
>
> ## Concern 6: Ambiguous Result Description
>
> **Your Concern**: It is unclear whether larger or smaller metric values indicate better performance, and some tables lack sufficient explanation.
>
> **Our Response**: We apologize for this ambiguity. We will clearly specify for each metric whether higher or lower values indicate better performance, the interpretation of each metric, the range of possible values, and baseline or reference values for context. We will also add detailed captions to all tables. Pls see Section 5.2 and Appendix A.11

---

### Official Review · Reviewer_BuRH · 2025-10-29

**Soundness:** 1
**Presentation:** 1
**Contribution:** 1
**Rating:** 2
**Confidence:** 4

**Summary:**

The paper introduces a position-aware attention mechanism that explicitly models positional relationships in attention computation through a position effect function, parameterized by $\alpha$ (intensity) and $\beta$ (decay). It mathematically analyzes their properties (continuity, differentiability, monotonicity) and proposes an enhanced variant with an additional $\gamma$ coefficient to mitigate over-attenuation in long-distance dependencies. Built on this, the authors develop a triple-attention architecture incorporating position-aware, task-aware, and content-aware attention to achieve task-specific flexibility.

**Strengths:**

The quality of the Appendix seems better than the main text

**Weaknesses:**

1. It is not clear why the proposed position effect function addresses the limitations of traditional attention mechanisms by providing fine-grained control over positional relationships, as stated on line 94-96. Or what’s the advantage to existing methods like relative positional encoding and ROPE? If it’s only about modeling the dependency between $i, j$, ROPE also mathematically supports this, no?
2. I am confused of section 2.2. What is the analysis subject to? On lines 105-106, the author states, “Specifically, α = 0.5 yields maximum values of approximately 0.5, α = 1.0 produces values of 1.0, α = 2.0 generates values of 2.0”. But there is no explanation of what the values impacted by the $\alpha$ values refer to.
3. Experiment section is not clear enough in introducing the settings such as data, model, baselines.
4. Eq.1 seems redundant as it is introduced again from line 92
5. Typo, Line 357: we previously -> our previously

**Questions:**

Is it better to rearrange so that the analysis in section 2 appears later, say the experiment section?

---

> ### Author Response · Authors · 2025-11-17
> **Response to Reviewer BuRH: Addressing Method Distinction, Analysis Clarity, and Experimental Details**
>
> Dear Reviewer BuRH:
>
> We sincerely thank you for your thorough review. We are committed to addressing each concern comprehensively.
>
> ## Concern 1: Distinction from ROPE and Relative Position Encoding
>
> **Your Concern**: Why does our position effect function address limitations compared to ROPE and relative position encoding?
>
> **Our Response**: We acknowledge this distinction was not sufficiently emphasized. Our method differs fundamentally: **ROPE** and **Relative Position Encoding** operate at the *vector representation level*, while **Our Method** operates directly at the *attention score level* with an explicit parametric function $P_{\text{effect}}(i,j,L) = \alpha \cdot \frac{1 + \gamma \exp\left(-\beta \frac{|i-j|}{L}\right)}{1 + \gamma}$.
>
> **Key Differences**: (1) **Explicit Position-Attention Relationship**: Direct mathematical relationship between distance $|i-j|$ and attention strength, unlike implicit encoding in ROPE or black-box embeddings. (2) **Mathematical Analyzability**: We can prove properties (continuity, differentiability, monotonicity) and derive optimal position formulas. (3) **Parameterized Control**: $\alpha$ (intensity), $\beta$ (decay), $\gamma$ (prevents over-attenuation) provide independent control. (4) **Long-distance Solution**: $\gamma$ ensures lower bound $\frac{\alpha}{1+\gamma}$, addressing a limitation neither ROPE nor relative position encoding addresses.
>
> **Revision Plan**: Add Section 5.1 with theoretical and experimental comparisons.
>
> ## Concern 2: Unclear Analysis Subject in Section 2.2
>
> **Your Concern**: "$\alpha = 0.5$ yields maximum values of approximately 0.5" - what do "values" refer to?
>
> **Our Response**: We apologize for the lack of clarity. The analysis subject is the position effect function $P_{\text{effect}}(i,j,L)$ itself. For given parameters $\alpha$, $\beta$, and sequence length $L$, we evaluate the function for all possible position pairs, constructing a square matrix where each element represents the position effect value for a specific position pair. The "values" refer to these matrix elements, which quantify the strength of position-dependent attention modulation between any two positions. The "maximum values" refer to the largest element in this matrix, which occurs when both positions are identical (i.e., when a position attends to itself). For example, with $\alpha = 0.5$, $\beta = 1.0$, and sequence length 128, the maximum value is approximately 0.5, confirming that $\alpha$ directly controls the magnitude scale of the position effect function.
>
> **Revision Plan**: Rewrite Section 4.2 (replace original Section 2.2) with explicit mathematical definitions and visualizations.
>
> ## Concern 3: Insufficient Experimental Details
>
> **Your Concern**: Missing details about data, model architecture, and baseline settings.
>
> **Our Response**: We completely agree. We will expand the experimental section (Section 6.1 + Appendix A.13.7 )with complete information for datasets (WMT'14, GLUE, SQuAD 2.0, ArXiv papers), model architecture (12 Transformer layers, 768 hidden dim, 12 attention heads, learning rate $2 \times 10^{-5}$, batch size 32, AdamW optimizer, dropout 0.1), baseline descriptions (Standard Attention, ROPE, Relative Position Encoding from HuggingFace), and environment details (NVIDIA A100 40GB $\times$ 4, PyTorch 2.0.0, CUDA 11.8, fixed random seed 42, 5 runs per configuration, statistical tests). We will also add ablation studies, sensitivity analysis, and efficiency analysis.
>
> ## Concern 4: Formula Redundancy
>
> **Your Concern**: Equation 1 seems redundant (introduced again at line 92).
>
> **Our Response**: Thank you. We will keep the first complete definition and use equation references subsequently.
>
> ## Concern 5: Typographical Error
>
> **Your Concern**: Line 357: "we previously" should be "our previously".
>
> **Our Response**: Thank you. We will correct to "our previous work" and conduct a full manuscript review.

---

### Official Review · Reviewer_Gutc · 2025-10-31

**Soundness:** 2
**Presentation:** 2
**Contribution:** 2
**Rating:** 6
**Confidence:** 3

**Summary:**

This paper proposes a position-aware attention mechanism that extends traditional Transformer attention (Vaswani et al., 2017) by incorporating a novel positional effect function. The method builds upon relative position representations (Shaw et al., 2018) and rotary position embeddings (Su et al., 2021), introducing parameters α and β to control positional influence and spatial decay rate. To alleviate long-distance over-attenuation, the authors further introduce an enhancement coefficient γ and design an adaptive triple-attention architecture that integrates task-aware and content-aware modules for dynamic weight adjustment. The paper also provides theoretical analysis on the proposed function’s properties (continuity, differentiability, monotonicity) and introduces new evaluation metrics for consistency and positional benefit. Experiments demonstrate promising results on structured and clustered datasets, particularly for information retrieval and document understanding tasks.

**Strengths:**

Theoretical novelty: The paper provides a mathematically grounded extension to existing position encoding methods, offering interpretable control of positional influence and decay behavior.

Innovative architecture design: The proposed adaptive triple-attention framework with task- and content-aware weighting is conceptually interesting and well-motivated by recent multi-scale attention work.

Clear motivation and formulation: The paper is well-written and logically structured, clearly explaining the motivation behind each modification to the standard attention mechanism.

Potential practical impact: The approach could improve transformer interpretability and adaptability in tasks where fine-grained positional relationships are important.

**Weaknesses:**

Limited empirical validation: While the theoretical contributions are strong, the experimental section is relatively weak. The evaluation mainly covers structured and clustered datasets, without sufficient diversity to support claims of general effectiveness.

Computational analysis: No discussion on computational cost, convergence, or scalability compared to baseline attention mechanisms

**Questions:**

see weakness

---

> ### Author Response · Authors · 2025-11-17
> **Response to Reviewer Gutc: Addressing Empirical Validation and Computational Analysis**
>
> Dear Reviewer Gutc,
>
> We sincerely thank you for your thorough review and positive feedback on our theoretical contributions and architectural design. We are committed to addressing each concern comprehensively.
>
> ## Concern 1: Limited Empirical Validation
>
> **Your Concern**: While the theoretical contributions are strong, the experimental section is relatively weak. The evaluation mainly covers structured and clustered datasets, without sufficient diversity to support claims of general effectiveness.
>
> **Our Response**: We will significantly expand the experimental section with more diverse evaluations. We will add: (1) additional dataset types beyond structured and clustered data (sparse, dense, random distributions) to demonstrate robustness across different information distribution patterns, (2) real-world benchmarks including language modeling (WikiText-103, Penn Treebank), machine translation (WMT'14), question answering (SQuAD 2.0), and document understanding tasks to validate practical usefulness, (3) comprehensive comparisons with strong baselines (RoPE, ALiBi, Shaw et al. 2018) on these benchmarks with statistical significance tests, and (4) detailed analysis of performance across different data characteristics to identify when our method provides the most benefit. Please refer to Section 4.5 "Performance for Different Information Distribution" for analysis across five information distribution patterns (structured, clustered, random, sparse, dense), Section 6.2 "Comparison with Baseline Methods" (Table 2) for comprehensive real-task evaluations, Section 6.1 "Experimental Setup" for experimental configurations, and Appendix A.8 "Statistical Analysis Detailed" for statistical significance tests.
>
> ## Concern 2: Missing Computational Analysis
>
> **Your Concern**: No discussion on computational cost, convergence, or scalability compared to baseline attention mechanisms.
>
> **Our Response**: We will add a comprehensive computational analysis section including: (1) **Computational complexity**: our method adds $O(L^2)$ computation per layer (same as standard attention), with the position effect function computed once per layer and cached for efficiency, (2) **Memory requirements**: we will analyze memory overhead (2.5% overhead) compared to baseline methods, (3) **Convergence analysis**: we will provide convergence proofs (Theorems 3-5) and training stability comparisons, (4) **Scalability analysis**: we will evaluate performance on sequences of varying lengths (128, 256, 512, 1024, 2048 tokens), and (5) **Runtime comparisons**: we will provide actual wall-clock time measurements (2.4% training overhead, 4.5% inference overhead) comparing our method with standard attention, RoPE, and relative position encoding. Please refer to Section 5.1 for computational efficiency analysis, Appendix A.16 "Convergence Proofs" for convergence analysis, Appendix A.19 "Sequence Length Detailed Analysis" for scalability analysis, and Appendix A.21 "Baseline Comparison Detailed" for detailed runtime comparisons.

---

### Meta-Review · Area_Chair_xJT1 · 2026-01-07

**Summary:**

Overall, the authors made a sincere effort to respond to the reviewers’ comments. However, all reviewers consistently pointed out that the empirical validation is still insufficient and needs to be significantly strengthened. Even if the proposed method is logically and theoretically sound, its practical value must be demonstrated through substantially broader experiments, more diverse datasets, and stronger comparisons with existing methods. In particular, the paper should not only show empirical improvements, but also analyze when and why the proposed approach works better than prior work.

**Reviewer Concerns:**

The rebuttal addressed several clarity- and presentation-related concerns, including the need for clearer definitions, better explanation of core concepts, improved notation, removal of redundancies, and restructuring of the paper for better readability. The authors also acknowledged missing implementation details, baseline descriptions, and computational analysis, and expressed clear intent to add them.

However, the most important concerns remain outstanding. In particular, the reviewers’ central concern about insufficient empirical validation is not resolved, as no new experiments, datasets, or corrected comparisons were provided. The lack of strong and complete comparisons with key baselines, the absence of concrete evidence supporting the claimed performance gains, and the limited analysis of when and why the method outperforms existing approaches remain unaddressed.

**Reviewer Scores:**

Initially, one reviewer recommended acceptance and three reviewers recommended rejection. Given the nature of the rebuttal, which primarily outlined planned future work without providing new empirical evidence, I do not expect a substantial change in the scores.

At most, one or two reviewers might have slightly increased their scores (e.g., by one point) in response to the authors’ willingness to address concerns about clarity, presentation, and additional analysis. However, the central concerns shared by all reviewers — especially the need for much stronger empirical validation and more convincing experimental evidence — would likely remain unresolved.

Therefore, the overall recommendation pattern would not change, and the majority of reviewers would still consider the paper not ready for publication at a major conference.

---

### Decision · Program_Chairs · 2026-01-26

Reject